# Multifunctional Performance of a Nano-Modified Fiber Reinforced Composite Aeronautical Panel

**DOI:** 10.3390/ma12060869

**Published:** 2019-03-15

**Authors:** Maurizio Arena, Massimo Viscardi, Giuseppina Barra, Luigi Vertuccio, Liberata Guadagno

**Affiliations:** 1Department of Industrial Engineering, Aerospace Section, University of Naples “Federico II”, Via Claudio 21, 80125 Naples, Italy; Maurizio.arena@unina.it; 2Department of Industrial Engineering, University of Salerno, Via Giovanni Paolo II, 84084 Salerno, Italy; lvertuccio@unisa.it

**Keywords:** nanocomposite, carbon fiber reinforced composite, damping, carbon nanotubes, laminate

## Abstract

The adoption of multifunctional flame-resistant composites is becoming increasingly attractive for many components of aircrafts and competition cars. Compared to conventional alloy solutions, the reduced weight and corrosion resistance are only a couple of the relevant advantages they can offer. In this paper, a carbon fiber reinforced panel (CFRP) was impregnated with an epoxy resin enhanced using a combination of 0.5 wt% of carbon nanotubes (CNTs) and 5 wt% of Glycidyl-Polyhedral Oligomeric Silsesquioxanes (GPOSS). This formulation, which is peculiar to resins with increased electrical conductivity and flame-resistance properties, has been employed for manufacturing a carbon fiber reinforced panel (CFRP) composed of eight plies through a liquid infusion technique. Vibro-acoustic tests have been performed on the panel for the characterization of the damping performance, as well the transmission loss properties related to micro-handling treatments. The spectral excitation has been provided by an acoustic source simulating the aerodynamic pressure load agent on the structure. The incorporation of multi-walled carbon nanotubes MWCNTs in the epoxy matrix determines a non-trivial improvement in the dynamic performance of the laminate. An increased damping loss factor with reference to standard CFRP laminate and also an improvement of the sound insulation parameter was found for the specific test article.

## 1. Introduction

More than one-fifth of the European Gross Domestic Product (GDP) is linked to the development and marketing of materials, components, technologies, and processes. Starting from this data, research into advanced materials—enabling technology according to the EU 2020 strategy—intends to create new and cheaper substitutes for existing materials and to develop new products and services with a high added value in areas such as health, aerospace, transport, and energy. In this contest, multifunctional materials play a prominent role. These materials are created by intimately integrating different types of materials as well as functions for implementing new applications that can be both technically challenging and economically favorable [1,2,3,4,5,6]. The combination of different materials may result in new functionalities not present in either of the single materials and, thanks to this specific ability to provide new levels of functionalities, multifunctional materials are promising, adaptable, and tailorable for future engineered systems [7,8,9,10,11,12]. Multifunctional materials allow saving in the number of parts and hence a lightweight design can be easily achieved with the consequent reduction of costs and resources. Besides this, by integrating the function of two or more different components, these composites are able to enhance the total system efficiency. Polymer nanocomposites constitute a relevant class of multifunctional materials. There are more general composites, having a filler with at least one of its dimensions at a nanoscale level, that are able to improve and/or modify the properties of a polymeric material to meet specific requirements [13]. Carbon-based nanofillers such as graphene, carbon nanotubes (CNTs), and carbon nanofibers (CNF) present astonishing mechanical properties but are also able to confer specific functionalities to meet pressing industrial requirements [14]. In general, the distinctive high-performance properties of carbon nanoparticles allows their use in many different applications [15,16]. CNTs have attracted attention mainly because of their size, shape, and high electrical conductivity [17]. Compared to conventional filler, CNTs are characterized by a high aspect ratio which, even at very low loading, increases the probability for percolation and conductive pathways throughout the matrix of the material [18,19]. Consequently, the addition of very low contents of CNTs is enough to impart an enhancement on the physical and chemical properties of the composite material, with the benefits of maintaining all other desired properties of the hosting matrices and their recyclability [20]. The use of carbon nanoparticles to improve the electrical properties of polymer nanocomposites provides a high potential for enhancements in electrostatic dissipation, printable circuit wiring, transparent conductive coatings, and electromagnetic interference (EMI) shielding [21,22,23]. Furthermore, the inclusion of CNTs within a polymer matrix can enhance not only the electrical properties but also other important properties of the material such as mechanical and thermal properties, gas barrier efficiency, and damping [24,25,26,27]. Compared to the neat polymer matrix, composite strength, energy-absorption, and working temperature range can be potentially increased [28]. Therefore, careful engineering of the carbon nanofiller type, the condition of processing, and the amount to load in the polymer matrix allows tailoring of the polymer nanocomposite to the specific application. High-density materials used because of specific electrical or thermal properties can therefore be replaced with lower density engineered polymer nanocomposites. Finally, the use of polymer nanocomposites allows for a reduction of the weight of high-performance components with an increase in the efficiency and lifetime of the final polymeric product. The weight reduction of components of primary structures has a major impact in the transport field [29]. In fact, transport is currently responsible for about one-fifth of the global CO_2_ emissions, and nearly a third of transport-related CO_2_ emissions originate from urban passenger transport. It also contributes to at least half of the air pollution in cities, seriously affecting the health of its inhabitants. The fuel reduction as a consequence of the weight reduction of vehicles directly results in a reduction of CO_2_ emissions and pollution [30]. Polymeric nanocomposites have also been proven to have a positive impact on photooxidation processes and corrosion prevention of electronic components or composite structural parts in the transport field with consequent increase of the service life of the component, and reduction of the cost for maintenance [31,32].

Composites are very sensitive to fire, which is a limiting issue in many applications. Literature has faced this issue [33], considering a full simulation of the composite structure to analyze the fire behavior, but in all cases the material has driven the attention of the investigation in the direction of increased performance as the main active constituent, which would be the matrix not only for degradation features but also for mechanically induced damages and failure. Many papers have dealt with the possibility of fire resistance filler, which could act twofold by increasing the fire resistance and suppressing the smoke release [34,35], and different materials with specific dimensionality have been considered and analyzed. In the case of fire performance, nanotechnology could be a viable tool to enhance the composite performance. In a previous paper [36], Raimondo et al. used the addition of POSS and CNTs to improve fire performance and synergically obtain multifunctional epoxy materials. In this paper, the authors have shown that the addition of different types of POSS may result in a dramatic increase of the limiting oxygen index (LOI) and a non-negligible reduction of the heat release rate (HRR) measured at the cone calorimeter. Other more recent papers [37,38] have presented the mechanical and morphological characterization of multifunctional carbon fiber-reinforced composites (CFRCs) prepared with nanocomposite epoxy matrices containing both POSS and CNTs.

As said before, in the aeronautic field, advanced composite materials have contributed to measurable improvements in weight savings, maintainability, durability, and reliability, but with poor acoustic performance. In fact, for aircraft skin panels, a continuous range of critical frequencies (critical region) is determined by the fact that the bending stiffness of the ply changes gradually from the softest direction to the stiffest direction. For composite materials, the high stiffness to weight ratio means that this region is at lower frequencies than for metallic panels and it falls just in the range where the human hearing mechanism is more sensitive.

Several approaches have been proposed to improve the damping performance in composite materials [39,40]. For instance Zarrelli et al. [40] explored the possibility of enhancing the damping performance of composite materials by micromechanical hybridization of fiber reinforcement. Their work presented an interesting investigation on the effect of viscoelastic fiber located within the carbon reinforcement architecture, which led to a significant improvement of the longitudinal and transverse damping coefficient. The major drawback in this case would be the unwanted reduction of elastic properties of the overall final composite. Previous research also demonstrated that noise and vibration control in aerospace composite structures can be favored with the use of nanotechnology [41,42,43,44,45]. This paper aims to develop structural composite materials with embedded integrated acoustic dumping functionality. The set goals have been achieved through the application of an advanced background on nanotechnologies for the development of new structural materials, based on the inherent ability of nanoscale reinforcement, where at least one dimension is limited to 1–100 nanometers (e.g., carbon nanotubes) to increase the energy dissipated by the material by three orders of magnitude. Nanostructured forms of carbon are particularly strong materials characterized by C–C covalent bonding and continuous hexagonal network architecture. A mechanism for damping enhancement in CNT composites [46] was first proposed by Rajora and Jalili [47]. They proposed the concept of the Stick-slip mechanism arising from the peculiar strong mechanical characteristics of CNTs. According to this concept, if a nanocomposite polymer is under applied stress, the load is transferred from the polymer to nanotubes causing both the matrix and the nanoparticles to accordingly deform until a certain value of stress is reached (critical shear stress). After this value, the nanotube debonds from the polymeric matrix which continuously deforms while the strain in the nanotube keeps staying constant. In this phase, called the slipping phase, no more load transfer occurs, and energy dissipation due to the slippage between the polymeric matrix and the nanotubes causes the damping of the structure [48]. This paper hence describes an attempt to investigate the response of the materials used as acoustic insulation coating embedded in the fuselage of the aircraft. The effects of carbon nanotubes and polyoctahedral silsesquioxanes (POSS), and in particular of glycidyl oligomeric silsesquioxanes (GPOSS) on a Tetraglycidyl methylene dianiline (TGMDA) epoxy formulation—free from any toughening or other additives, normally used in commercial resin to improve the viscoelastic response of the material, such as elastomers—have been investigated. The choice of using a blank formulation has been made in order to discriminate and highlight only the effects of the above-mentioned components.

## 2. Materials and Methods

### 2.1. Materials

Epoxy resin. The epoxy precursor was prepared using Tetraglycidyl methylene dianiline TGMDA (Epoxy equivalent weight 117–133 g/eq) as the epoxy monomer and 1,4-butanedioldiglycidylether (BDE) acting as the reactive diluent, obtained from Sigma–Aldrich (Milan, Italy). The epoxy monomer and the reactive diluent were mixed together at a ratio of 75:25 wt% 

Carbon nanotubes. The multi-walled carbon nanotubes MWCNTs (3100 Grade) were obtained from Nanocyl S.A. (Sambreville, Belgium). MWCNTs (3100 Grade) had the following characteristics: Outer diameter varying from 10 nm to 30 nm; length varying from hundreds of nm to some micrometers; number of walls ranging from 4 to 20; specific surface area 250–300 m^2^/g; and the carbon purity was >95% with a metal oxide impurity <5%. An amount of 0.5 wt% of MWCNT was used for blend preparation.

POSS molecules. GPOSS was purchased from *Hybrid Plastic* (Hattiesburg, MS, USA) and used to prepare POSS/epoxy composites with 5 wt% of POSS.

Carbon fibers. Plain weave Carbon fabric was used for the preparation of the composites. In particular, HEXCEL HexForce^®^ G0814 6 1000 TCT Carbon Fabric (HEXCEL, Saronno (VA) Italy) with an areal density of 0.193 kg/m^2^. The thickness was 0.2 mm.

Curing agent. 4,4′ diaminodiphenyl sulfone (DDS) purchased from Sigma–Aldrich (Milan, Italy) was used as a hardener agent and added at a stoichiometric concentration with respect to all the epoxy rings arising from TGMDA, BDE, and POSS.

#### 2.1.1. Preparation of the Unfilled Epoxy Matrix (Epoxy)

Epoxy blend (TGMDA and BDE) and DDS were mixed at 120 °C until the complete solubilization of the hardener. Afterwards, the mixture was cooled to 90 °C.

#### 2.1.2. Preparation of the Epoxy Nanofilled Matrices

Epoxy blend (TGMDA and BDE) and DDS were mixed at 120 °C until complete hardener solubilization and then the mixture was cooled to 90 °C. At this temperature Carbon nanotubes and/or GPOSS compounds were added using ultra-sonication for 20 min. Hielscher model UP200S (200 W, 24 kHz) (Hielscher Ultrasonics GmbH, Teltow, Germany) ultrasound was used for this last step. Table 1 indicates the name and formulation of all the prepared epoxy matrices.

#### 2.1.3. Manufacturing of the Carbon Fiber Reinforced Panel (CFRP)

CFRCs were obtained using a liquid infusion technique for the formulation GP-CNT. The vacuum bag was prepared according to the procedure described elsewhere [49]. A laminate configuration [0,90]_4_ was used for the purpose. A scheme of the vacuum bag for the liquid infusion is shown in Figure 1. With this procedure, the resin was forced to flow through the thickness of the preform using an external vacuum pump. After sealing, the vacuum bag was transferred into the autoclave for curing, pressure and temperature conditions are shown in Figure 2. Figure 3 shows the final manufactured panel.

The non-traditional liquid infusion technique for the preparation of the carbon fiber reinforced polymer (CFRP) panels has already been used by Guadagno et al. [49,50] to obtain CFRP panels impregnated using resin with 0.5 wt% CNTs and turned out to be successful in reducing filtration problems and the formation of big CNTs agglomerates. The authors demonstrated that with this unconventional technique, compared to the traditional resin infusion process, significant improvements of the electrical conductivity were achieved, equal respectively to about 72% for the in-plane and 120% for the out-plane. Such outstanding improvement was associated with the mechanisms governing the infusion process. In particular, morphological analysis on the sections of etched panels demonstrated that the difference in the electrical conductivity was closely related to the distribution of CNTs between the carbon fiber (CF) plies. The authors found that, in the case of traditional Resin Transfer Molding (RTM) techniques, the network of carbon nanotubes was preferentially arranged between the plies of the carbon fibers in directions approximately parallel to the plane of the panel, whereas in the case of bulk liquid infusion CNTs were more copiously arranged through the section of the panel perpendicularly aligned to the ply planes.

### 2.2. Methods

A Static mechanical test, Dynamic mechanical analysis (DMA), Thermogravimetric analysis (TGA), and LOI tests were performed on cured epoxy matrix. Samples were prepared by transferring the epoxy filled and unfilled matrices—molds with opportune dimensions to prepare the necessary coupons for these tests. The curing cycle which was performed in an oven at atmospheric pressure and consisted of a 1 h initial step at 125 °C followed by a 3 h second step at 180 °C. Vibro-acoustic characterization was performed on the carbon fiber reinforced composite.

#### 2.2.1. Static Mechanical Test of Cured Epoxy Matrix

Static mechanical tests were performed according to ASTM D638-14 standard (Standard Test Method for Tensile Properties of Plastics, 2014) (see Figure 4). Five samples were tested for each compound in the tensile axial loading at a rate of 1 mm/min. Young’s modulus was obtained from measuring the slope of the stress–strain curve in the linear region. The results were reported as the average value between the 5 performed tests for each sample and the standard deviation as a measurement of the error.

#### 2.2.2. Dynamic Mechanical Tests

A thermo-analyzer (Tritec 2000 DMA -Triton Technology, Mansfield, MA, USA) was used to perform dynamic mechanical tests on coupons with dimensions 2 × 10 × 35 mm^3^. A variable flexural deformation in three points bending mode, with an amplitude of 0.03 mm at the frequency of 1 Hz was applied. Samples were scanned at 3 °C/min from −90 °C to 315 °C.

#### 2.2.3. LOI Tests

Limiting oxygen index (LOI) was measured according to the standard ASTM 2863 on all the samples. Specimens with dimensions of 80 × 10 × 3 mm^3^ were fixed in a vertical position within a tube in an atmosphere where the relative concentration of oxygen and nitrogen could be changed. A small pilot flame was applied from the top to allow the specimen to burn downwards in a candle-like manner, at the opportune oxygen concentration. In this condition, the minimum oxygen concentration required just to sustain combustion of the sample was determined and expressed as:(1)$$LOI=\frac{\left[O_{2}\right]}{\left[O_{2}\right]+\left[N_{2}\right]}\times 100.$$

#### 2.2.4. Thermogravimetric Analysis (TGA)

A Mettler TGA/SDTA 851 thermobalance (Milan, Italy) was used to perform thermogravimetric analysis (TGA). The analysis, in which the weight loss was recorded as a function of the temperature, was carried out using a temperature range from 25 °C to 900 °C at a 10 °C min^−1^ heating rate in both an air and nitrogen atmosphere.

#### 2.2.5. Temperature Programmed Oxidation (TPO)

Temperature programmed oxidation (TPO) experiments were performed using the apparatus assembled in the laboratory and further described below. Approximately 1 g of the sample (chips of about 1 mm shape) was heated under air flow (500 N cm^3^ min^−1^) from room temperature to 900 °C at a heating rate of 10 °C min^−1^. The experimental plant had a classical configuration with a feed, reaction, and analysis section. Air was fed to the reaction section by means of a mass flow-controller for gas (supplied by Bronkhorst High-Tech, Ruurlo, The Netherlands). The sample was diluted with quartz and loaded inside a tubular quartz reactor. The sample was placed in the isothermal zone of the reactor realizing a fixed bed sandwiched between two quartz wool flakes, and the reactor was placed in a Proportional Integral Derivative PID-controlled electrical oven (Salvis Lab Vacucenter VC 50, Rotkreuz, Schweiz, Switzerland) for the heating. The gaseous products’ composition was continuously monitored by an online Nicolet Antaris IGS FT-IR multi-gas analyzer (Thermofisher-scientific, Waltham, MA, USA) while oxygen profile was followed by means of a Hiden Analytical HPR-20 Mass Spectrometer (Hiden Analytical, Warrington, UK).

#### 2.2.6. Vibro-Acoustic Characterization of the Carbon Fiber Reinforced Composite

The main purpose of the vibro-acoustic test was to estimate the modal parameters in a large spectral field. For this specific purpose, to increase the modal extraction frequency band the test article was subjected to a pressure excitation emitted by a flat-type sound wave generator placed inside a reverberating box—a random signal was generated in order to excite all the natural frequencies in the bandwidth of interest. The panel was arranged in a simply supported configuration and soft material sheets (i.e., polystyrene) were bonded on the edges in order to avoid any coupling mechanism among the plunge rigid motion and the elastic mode shapes. Next to the outlet surface, a scanning laser head (Polytech PSV 400, Polytec GmbH, Waldbronn, Germany) was positioned to measure the vibration velocity of the test article and consequently elaborate on the relative Operative Deflection Shapes (ODS) of the panel, avoiding any local mass loading effect and spatial positioning error that have many times been associated with the use of standard accelerometers. One microphone was positioned inside the reverberating box to measure the incident Sound Pressure Level (SPL) while a Pressure–Pressure (PP) probe was used to detect the sound intensity that goes beyond the panel. Figure 5 illustrates the test setup.

## 3. Results and Discussion

### 3.1. Characterization of the Epoxy Matrix

#### 3.1.1. Dynamic Mechanical Tests

Dynamic Mechanical tests were carried out on all the epoxy matrices indicated in the previous section. The variation of the storage modulus, E’, and the corresponding loss factor, tanδ, with temperature are shown in Figure 6a,b. All samples show a temperature with lower than 180 °C plateau regions at typical E’ values corresponding to the glassy state of a polymer. At temperatures above 180 °C, the neat epoxy resin and the sample containing only GPOSS are characterized by a single drop in the storage modulus and by the presence of a unique tanδ peak associated with the glass transition, centered at 260 °C and 255 °C respectively. Previous work has shown that the complete solubility of GPOSS in the epoxy resin and the presence of epoxy functionalities in the POSS compound allows the formation of a continuous matrix [51]. In the same range of temperatures, both the samples containing CNTs instead exhibit two falls in the storage modulus and two peaks in the tanδ. For these samples, another peak at a lower temperature was evident in addition to the one centered at about 255 °C, related to the main glass transition of the epoxy matrix. This phenomenon has already been found previously by Guadagno et al. [25,52]. The authors ascribed this second peak to the presence of an additional phase with different crosslinking density. According to their findings, the presence of the carbon nanofiller strongly influenced the structure of the epoxy matrix causing, during the curing, discontinuity in the crosslinking reaction with the consequent formation of a phase with a reduced curing degree characterized by a lower glass transition. The reduction of the intensity of the main peak in favor of the one at a lower temperature suggests that two phases can coexist in the resin.

#### 3.1.2. Tensile Tests

The stress–strain curves recorded during the tensile test are illustrated in Figure 7. Representative curves of all material formulations were selected. For all the samples, an almost linear behavior followed by a sudden brittle fracture without significant deformations is observable.

From the stress–strain diagrams, Young’s modulus, tensile strength, and elongation at break have been extracted, and the results are summarized in Table 2, expressed as the average values ± the standard deviation. The introduction of CNTs or GPOSS in the Epoxy formulation increases all the mechanical parameters of the resin, but GP-CNT samples, which contain both GPOSS and CNTs, are the one with the highest stiffness and strength. As already said before GPOSS is totally soluble in the epoxy resin used in this work [36]. Moreover, epoxy functionalities in GPOSS contribute to the matrix cross-linking reaction, allowing the formation of a continuous matrix system. Such a system will most likely make the load transfer between the POSS units inside the epoxy matrix possible, which in turn determines an increase in Young’s modulus. The increase of the values at breakpoint could be ascribed to an increase in the flexible chains in the matrix tracts due to the presence of linear side groups in the GPOSS cage. Because the epoxide network is brittle, the presence of soft dispersed-phase improves the strength at break. Compared to the neat epoxy resin, the synergistic effect of the GPOSS and CNTs allows us to, on the one hand, obtain an effective load transfer distributed on the units inside the epoxy matrix, as evidenced by the increase of the Young’s modulus and the maximum load at break, and on the other hand, result in an increase in the strain at break point that could be ascribed to an increase in the flexible chains in the matrix tracts due to the presence of linear side groups in the GPOSS cage.

#### 3.1.3. LOI Test, Thermogravimetric Analysis (TGA), Temperature Programmed Oxidation (TPO)

Limiting oxygen index (LOI) values determined for all the epoxy formulations are indicated in Table 3. The addition of only GPOSS strongly affects the flammability of the resin, in fact, the LOI increased from 27% of the neat epoxy resin to 33% for the sample containing only POSS. There is extensive work in the literature on the use of POSS compounds as fire retardants for epoxy polymers [36,53,54,55,56]. It is widely recognized that a ceramic protective layer is generated during POSS degradation. In this process, the migration exerts an important effect on surface accumulation. Similar to silicones, POSS have low surface energy and can move toward the surface where they can be easily converted to ceramic species, acting as a protective layer. During heating, the non-volatile silicon derivatives create silica residues that accumulate on the surface while the polymer decomposes. The resulting surface barrier hampers the mass and heat transportation between the condensed and gas phase, delaying the volatilization of decomposition products [57]. Moreover, the groups on the GPOSS surface permit the solubility within the epoxy matrices, in fact, GPOSS is fully epoxidized with glycidyl groups and hence compatible with both the epoxy precursor and reactive diluent. The addition of CNTs causes a quasi-negligible effect on the flammability of the resin. In fact, the sample containing only CNTs is characterized by a LOI of 28%, just slightly higher than the LOI of the neat epoxy resin. LOI for the sample containing both GPOSS and CNTs is 30%, a value which is in between the value for the neat sample and that of the sample containing only GPOSS. We suppose that the high thermal conductivity of the CNTs counterbalances the beneficial effect of the presence the GPOSS on the improvement of the flammability of the epoxy matrix.

The thermogravimetric curves under the nitrogen atmosphere of Epoxy and GP-CNT samples are shown in Figure 8a. For both samples, two main weight-loss stages are observable at about 350 °C and 650 °C, respectively. The first stage is compatible with the release of the first degradation products such as ester/ether components, aromatics, and carbonyl compounds while the second one corresponds mainly to the release of methane, arisen from the fragmentation of epoxy chains during the pyrolysis [58]. Although the degradation of the GP-CNT sample starts at slightly lower temperatures with respect to the blank epoxy sample, the first weight loss stage process proceeded at a slower rate and ended up at 450 °C with a considerably reduced weight loss of 55% compared to the weight loss of 65% for the blank Epoxy sample. After that, the degradation continued for both the samples with the second stage reaching 900 °C with a weight loss of about 80%. Figure 8b shows the results of the thermogravimetric analysis in air. In this case, for both samples, two main weight loss stages are clearly visible, centered respectively at about 420 °C and 550 °C. For both the samples, the first stage process ended up at 450 °C with a weight loss of about 50%. The second stage continued for both samples up to about 700 °C with a final weight loss of about 99% for the unfilled epoxy sample and of 95% for the GP-CNT sample.

Figure 9 shows the results from the Temperature Programmed Oxidation (TPO) experiment for the Epoxy and GP-CNT samples. The oxygen consumed is plotted as a function of the temperature as well as water, methane, carbon monoxide, and carbon dioxide released during the oxidation. For both the samples, at temperatures lower than 300 °C, a release of water was observable which was physically adsorbed or linked via hydrogen bonding in the samples. The oxidation of the Epoxy and GP-CNT samples started at 360 °C and 330 °C, respectively, as at these temperatures the oxygen concentration started to drop down from the initial value of 21% and contemporaneously the concentration of the released water started climbing steadily, peaking at 439 °C and 330 °C for the Epoxy sample and GP-CNT, respectively. This first oxidation stage also produces degradative products such as methane. The release of methane started at 400 °C for the Epoxy sample and at 340 °C for the GP-CNT. With the rising of the temperature, a rapid drop of the oxygen concentration, occurring at 490 °C for the Epoxy sample and 460 °C for the GP-CNT sample, indicated the development of further oxidative paths. At the same temperatures, carbon monoxide and carbon dioxide start to be released. The onset temperatures, as well as the temperatures at the peak and offset temperatures for the concentration of the oxygen and for the released compounds during the combustion, are indicated in Table 4. According to the data indicated in Table 4, there is clear evidence that the presence of the nanofillers in GP-CNT anticipates all the oxidative processes. Romano et al. studying the thermal conductivity of epoxy resins filled with MWCNT and hydrotalcite clay [59] found that the addition of CNTs in epoxy matrices increased, in a non-negligible way, the thermal conductivity of the material. In our TPO measurement conditions, and in particular with regard to the shapes of the sample loaded in the reactor (chips of dimension of mm), the thermal conductivity may be a very relevant parameter for the oxidation process, at least at temperatures lower than 600–700 °C.

### 3.2. Characterization of the CFRP Panel

#### 3.2.1. Dynamic Characterization of the Manufactured Panel

The laser tests carried out on the examined sample brought about a series of significant results regarding the modal-vibrational behavior, with respect to acoustic excitation in the frequency range between 0 and 200 Hz. These results were then post-processed in Matlab^®^ (Mathworks, Torino, Italy) environment. The frequency response (vibration velocity/input pressure level) presents a series of peaks, which represent a measure of the modal vibrations with respect to an acoustic input load. Specifically, the peaks can be associated with defined resonance frequencies typical of the panel modes, and depending on these peaks, it is possible to extract the specimen’s damping coefficient for that specific mode. In such a way, it has been possible to quantify an important vibro-acoustic property of the analyzed material, the ability to limit its deformations under the action of a relevant pressure load. Excluding the initial frequency range with the maximum peaks (up to about 60 Hz) representative of a rigid body behavior of the sample, several resonances have been acquired in the spectral range between 60–200 Hz. Modal damping has further been evaluated through the half bandwidth method showing a remarkable dissipative effect of the panel in correspondence of the elastic modes (Figure 10). The operative deflection shapes of the composite panels are shown in Figure 11 and the modal damping value are reported in Table 5. This behavior confirms relevant results assessed during previous studies conducted by the authors that already showed an increase of up to about 3.5% of modal damping compared to a standard epoxy formulation [44]. The achieved results are surely widely demonstrative of the nanotubes efficiency. Generally, the average damping factor of the carbon fiber composites is approximately 1%, as discussed elsewhere [60,61,62].

#### 3.2.2. Sound Insulation Characterization

The sound insulation characterization test used the same layout of the vibrational test as herein described. The sample was acoustically loaded through a speaker generating flat acoustic waves, and acoustic parameters were measured below and above the panel partition. According to the UNI EN ISO 15186-1 standard procedure, incident sound pressure level was measured inside the reverberating box as an evaluation of the incident acoustic energy. On the outer surface of the panel, the sound acoustic intensity was measured using a pressure–pressure probe or PP probe with a variable spacer (1, 2, and 5 cm) to guarantee a valid frequency band from 31.5 to 6300 Hz.

During the data acquisition, the probe was positioned vertically to the plane of the panels, so that the acoustic wave moves along the axis of the meter maximizing the calculation of the average of sound pressures, and the same probe has been moved slowly along the area inside the panel. According to the standard procedure, the average sound intensity was measured:(2)$${\bar{L}}_{in}=10log\left(\frac{1}{S_{m}}\overset{n}{\underset{i=1}{\sum }}(S_{mt})\cdot 10^{\frac{L_{int}}{10}}\right)$$

The availability of the incident SPL and the outcome of Sound Intensity has been possible with the evaluation of the Sound Insulation parameters, through the application of the following formulation:(3)$$R_{i}=L_{p}-6-\left(L_{in}+10\log\frac{S_{m}}{S}\right)$$
where:*L_p_* is the average sound pressure level inside the reverberating box*L_In_* is the average sound intensity level*S_m_* is the measuring grid total area*S* is the partition total area

The computed sound insulation is reported in Figure 12. As for many of these elements, the sound insulation parameter presents three main regions: The first governed by the element stiffness properties (characterized by a descending profile), the second mainly controlled by the mass properties (with a first order upward profile locally modified by normal modes effects), and a coincidence area where a rapid decrease of the insulation is measured before it starts to increase again (with a second order ascending profile).

In the absence of a neat reference panel, the sound insulation performance comparison was performed with respect to other literature data on similar panel configuration. In 1980, L.R. Koval [63], carried out deep studies on the transmission of airborne noise into an aircraft fuselage. Results indicated that the noise attenuation in a laminated composite shell did not seem to offer any particular benefit over an aluminum shell mainly due to the increased acoustic radiation efficiency of CFRP. Figure 12 also compares data from the present experimental campaign with literature data on very similar panel configurations [64] and highlights the important improvement of performances of the nanostructured sample. The direct comparison evidence an important increment of the sound transmission loss (STL) in the very low-frequency range (mainly due to the stiffness increment of the panel) with respect to the metallic plate, but also improved performance in the whole frequency range due to the effect of damping as demonstrated in previous works [65,66]. It can be, in fact, assumed that the resonant path for STL is attributed to the coupling of acoustic waves to free bending waves in the panel. It usually dominates the overall response around and above the panel coincidence frequency, where the acoustic wavelength is about the same as the structural wavelength, making the panel radiation more effective. In this frequency range, it is the damping loss factor that primarily controls panel vibration response and consequently the sound transmitted through the panel, so the higher the loss factor, the higher the STL.

## 4. Conclusions

In this paper, a carbon fiber reinforced panel (CFRP) was manufactured using a new formulated epoxy-based matrix and an innovative liquid infusion technique. The matrix was an epoxy resin enhanced using a combination of 0.5 wt% of carbon nanotubes (CNTs) and 5 wt% of Glycidyl POSS (GPOSS). This formulation has some peculiar characteristics of improved mechanical and flame-resistance properties as well as improved resistance to the thermal degradation, we found that:The introduction of the CNT and GPOSS fillers in the epoxy precursor synergically determines the formation of a fraction of the resin with a lower Tg, which represent a phase with greater mobility of chain segments, most probably closely linked to the filler. Moreover, it allows us to obtain, on the one hand, an effective load transfer distributed on the units inside the epoxy matrix, as evidenced by the increase of Young’s modulus and the maximum load at break, and on the other, an increase in the strain of breakpoint that could be ascribed to an increase in the flexible chains in the matrix tracts due to the presence of linear side groups in the GPOSS cage.The obtained LOI for Epoxy and GP-CNT are 27% and 30%, respectively, indicating that the simultaneous addition of GPOSS and CNT leads to a material with better flame properties.Although the thermal degradation of the GP-CNT system started at slightly lower temperatures with respect to the blank epoxy sample, the first stage process proceeded with a reduced rate and ended up with a considerably reduced weight loss of 55% compared with the weight loss of 65% for the blank Epoxy sample. Thermogravimetric analysis in air, showed that for both the samples with and without nanofillers, two main weight loss stages were clearly visible, centered respectively at about 420 °C and 550 °C. For both the samples, the first stage process ended up at 450 °C with a weight loss of about 50%. The second stage continued for both the samples up to about 700 °C with a final weight loss of about 99% for the unfilled epoxy sample and of 95% for GP-CNT sample.Results from the TPO experiment indicated that the presence of the nanofillers in GP-CNT anticipates all the oxidative processes. This effect has been ascribed to an increase in the thermal conductivity due to the presence of CNTs.

The CFRP panel is characterized in terms of vibro-acoustic performance in order to evaluate both the damping and the transmission loss properties. The experimental results confirmed an increment of the overall damping factor of the specimen due to the simultaneous incorporation of CNT and GPOSS fillers. In fact, the measured modal damping reached doubled values with respect to typical CFRP epoxy panels. Regarding the sound transmission loss (STL), a comparison of data from the present experimental campaign with literature data on very similar panel configurations highlights the important improvement in performance of the nanostructured sample. An important increment of the sound transmission loss (STL) in the very low-frequency range (mainly due to the stiffness increment of the panel) with respect to the metallic plate has been determined, as well as improved performance in the whole frequency range due to the effect of damping.

## Figures and Tables

**Figure 1 materials-12-00869-f001:**
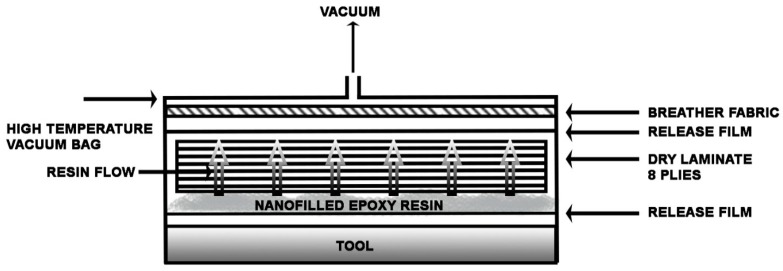
Vacuum bag preparation for the liquid infusion.

**Figure 2 materials-12-00869-f002:**
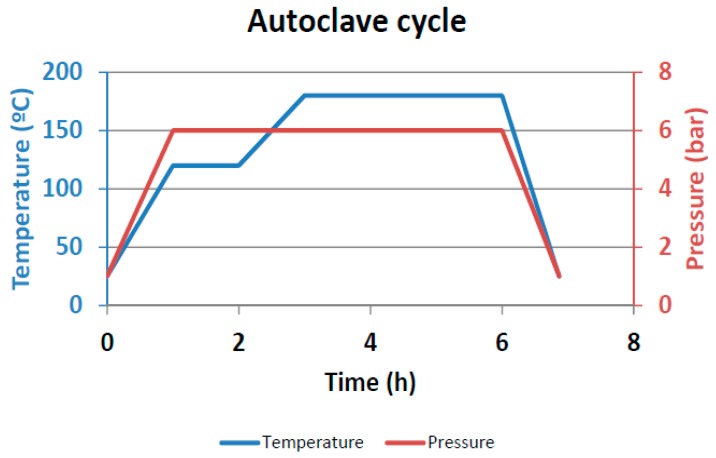
Curing cycle.

**Figure 3 materials-12-00869-f003:**
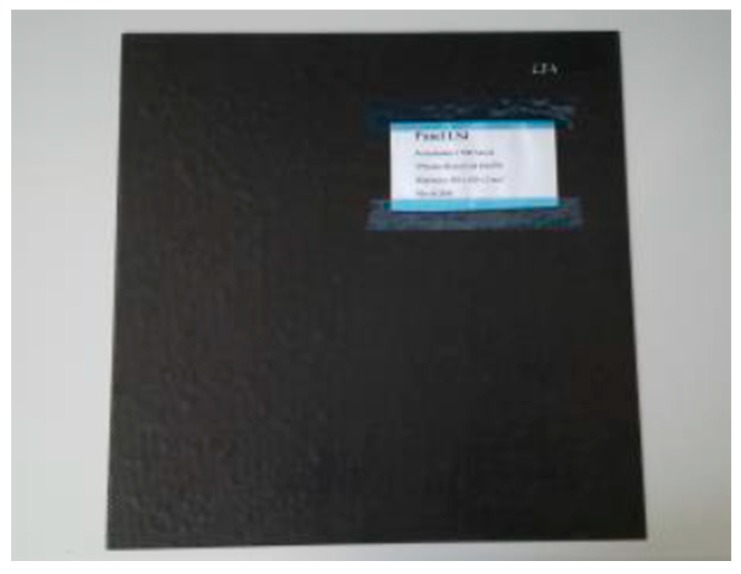
Manufactured panel.

**Figure 4 materials-12-00869-f004:**
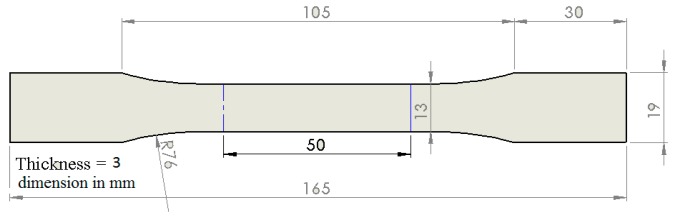
Specification of tensile test specimen according to ASTM D638-14. All dimensions in mm.

**Figure 5 materials-12-00869-f005:**
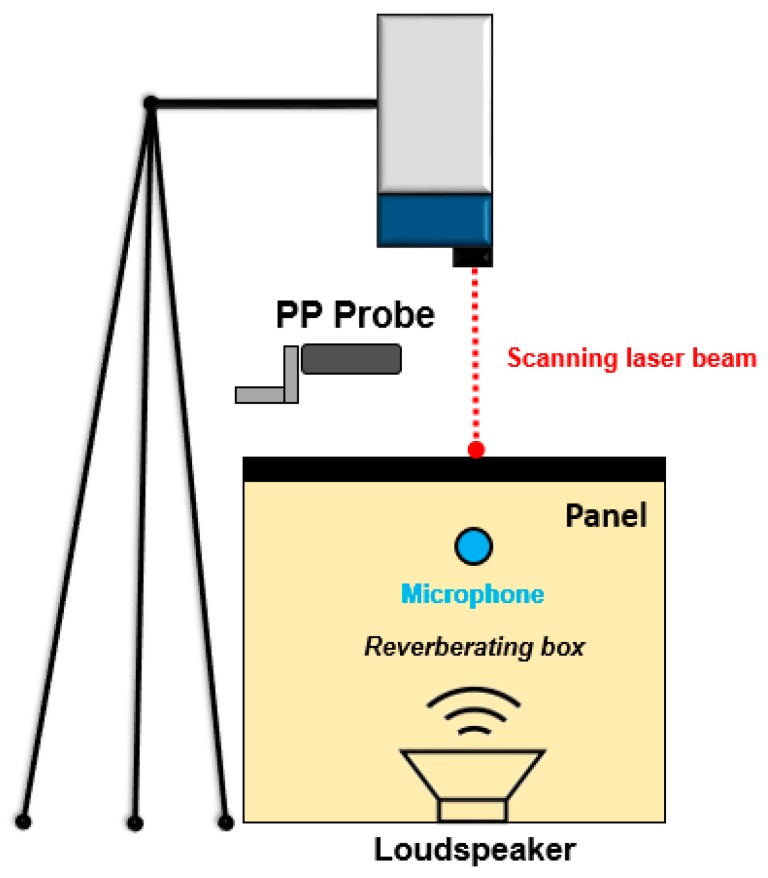
Vibro-acoustic characterization test setup.

**Figure 6 materials-12-00869-f006:**
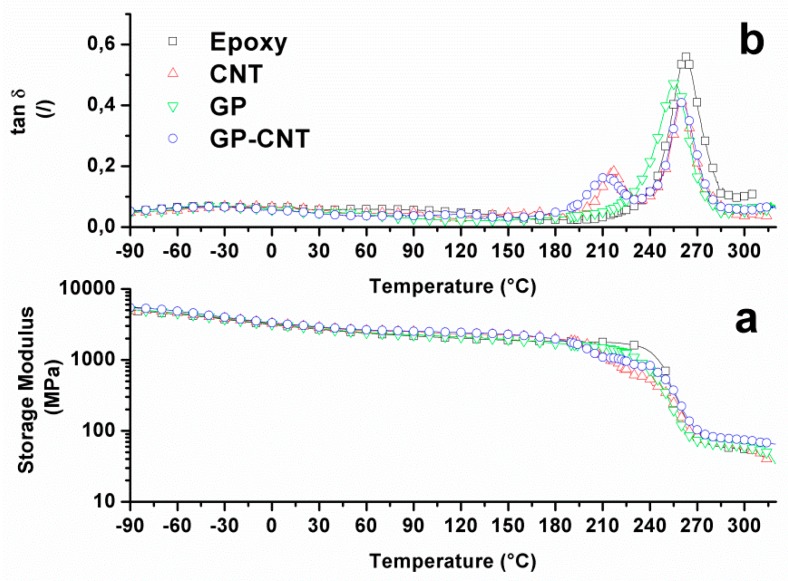
DMA analysis for the filled and unfilled epoxy resin (**a**) storage modulus (**b**) Loss factor (tanδ).

**Figure 7 materials-12-00869-f007:**
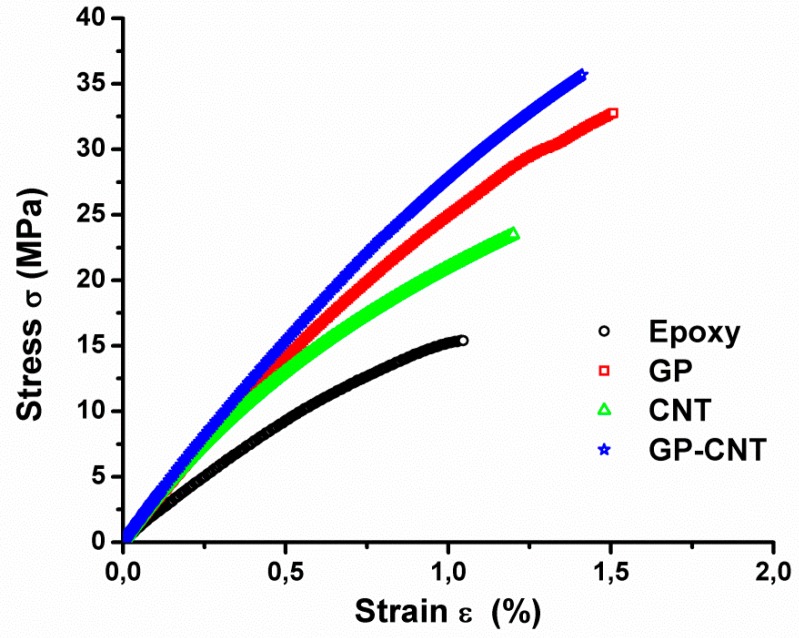
Stress–strain curves and mechanical parameters of epoxy resin with and without carbon nanotubes and GPOSS.

**Figure 8 materials-12-00869-f008:**
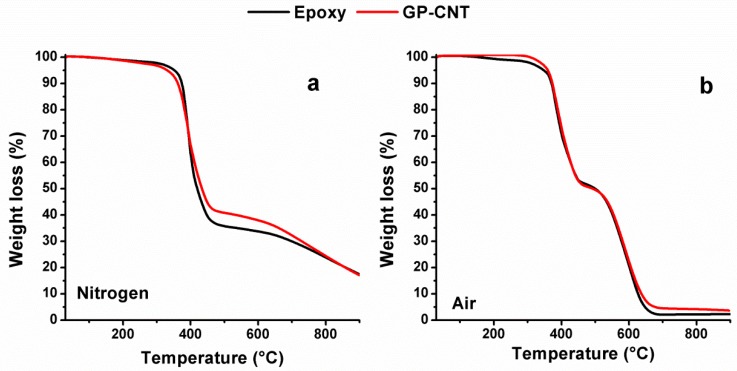
Thermogravimetry profile of epoxy resin with and without carbon nanotubes and GPOSS (**a**): in nitrogen atmosphere (**b**): in air.

**Figure 9 materials-12-00869-f009:**
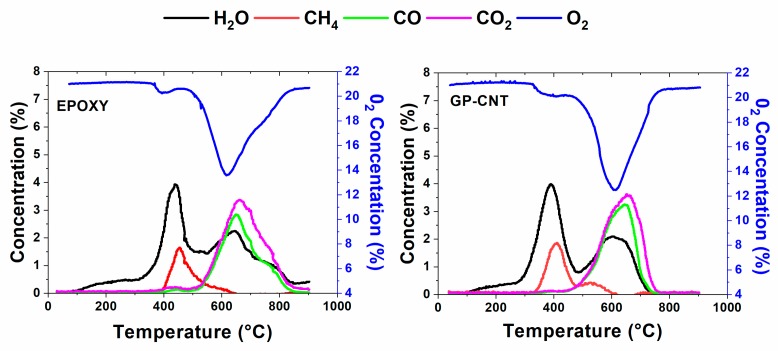
The oxygen consumed and gas released during oxidation in the temperature programmed oxidation (TPO) experiment.

**Figure 10 materials-12-00869-f010:**
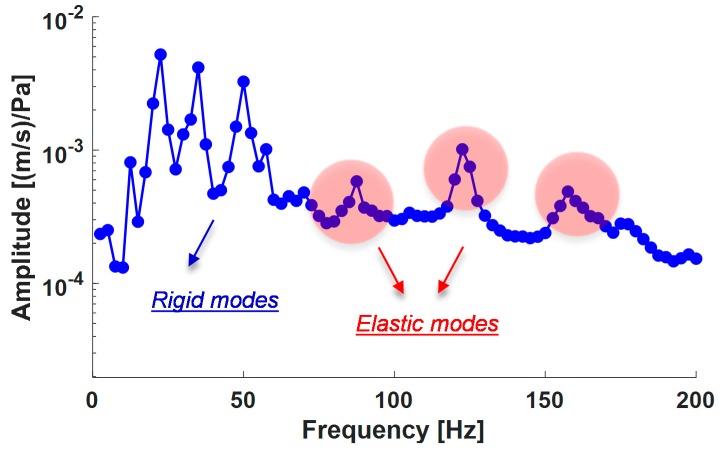
Frequency response function, laser vibrometry.

**Figure 11 materials-12-00869-f011:**
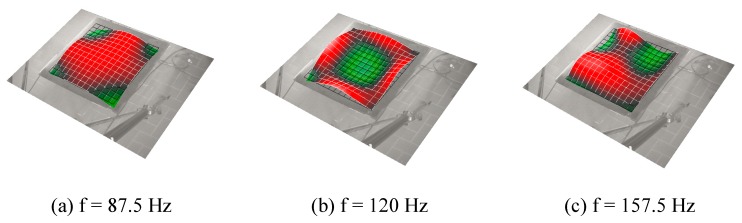
Operative deflection shapes of the composite panel.

**Figure 12 materials-12-00869-f012:**
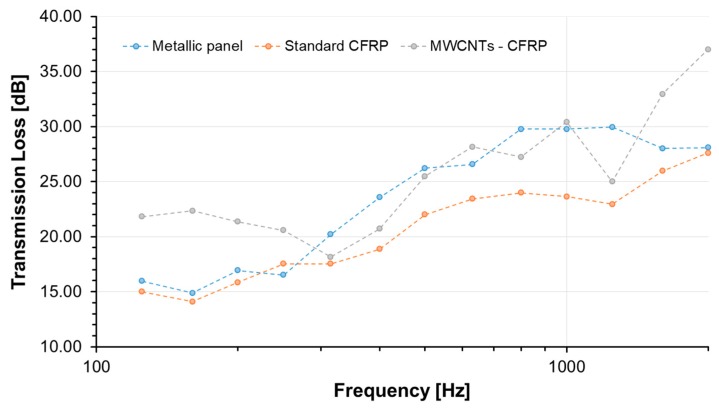
Transmission Loss: 1/3 Octave Bands Centre Frequency.

**Table 1 materials-12-00869-t001:** Epoxy matrices formulation names and MWCNTs and GPOSS contents.

Sample Name	CNTs (wt%)	GPOSS (wt%)
Epoxy	-	-
GP	-	5.0
CNT	0.5	-
GP-CNT	0.5	5.0

**Table 2 materials-12-00869-t002:** Young’s modulus, tensile strength, and elongation at break for filled and unfilled epoxy samples.

Sample	Young’s Modulus (Mpa)	Tensile Strength at Break (Mpa)	Elongation at Break (%)
Epoxy	2182.2 ± 120.5	17.3 ± 3.9	1.1 ± 0.1
CNT	3325.2 ± 80.3	22.2 ± 5.2	1.2 ± 0.1
GP	3076.5 ± 96.5	35.4 ± 4.8	1.6 ± 0.3
GP-CNT	3282.4 ± 62.7	38.1 ± 6.3	1.5 ± 0.2

**Table 3 materials-12-00869-t003:** Limiting oxygen index (LOI) for filled and unfilled epoxy samples.

Sample	LOI % ASTM 2863
Epoxy	27
GP	33
CNT	28
GP-CNT	30

**Table 4 materials-12-00869-t004:** Onset temperatures, temperatures at the peak, and offset temperatures for the concentration of the oxygen and for the released compounds during the combustion.

	Sample	T Onset (°C)		T Peak (°C)		T Offset (°C)
O_2_	EPOXY	360	491	610		852
	GP-CNT	330	460	618		770
H_2_O	EPOXY	330		439	645	835
	GP-CNT	290		388	610	750
CH_4_	EPOXY	398		455		644
	GP-CNT	340		411	536	615
CO	EPOXY	500		651		840
	GP-CNT	452		653		747
CO_2_	EPOXY	510		659		860
	GP-CNT	452		653		754

**Table 5 materials-12-00869-t005:** Modal damping values.

Resonance Frequency [Hz]	Modal Damping [%]
87.5	2.53
120	2.81
157.5	3.14

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
