# Peer review of "Multifunctional Performance of a Nano-Modified Fiber Reinforced Composite Aeronautical Panel"

_materials, 2019, doi:10.3390/ma12060869_

Round 1

Reviewer 1 Report

The topic of the paper. the damping, mechanical and flame retardant properties of CFRP is very interesting, however, the manuscript is not suitable for publication in the present form.

First of all, the English of the manscript needs to be improved.

The introduction is too general, it does not contain any data, and only a short part of it deals with the real content of the ms.

I suggest adding the informations about the EP curing cycle just after the description of the preparation.

Did the authors detect any sign of filtration of the CNTs during the resin injection process? Is there ny evidence that the CNT content is uniform throughout the composite?

In the case of the TGA measurements, the method description says that 100C min-1 heating rate was applied, please check it.

Please do not use expressions like next picture (e.g. line 204), refere instead to the number of the figure. (see also line 260, table xxx)

In section 3.1.1 the citation is missing for the mentioned previous work.

Moreover, the discussion about the double Tg in the case of the GP-CNT sample is not convincing.

I suggest the measurement of samples containing only GPOSS, and only CNTs as well, maybe the real explanation will show up.

Also in the case of tensile, TGA and LOI measurements, the results of the above mentioned samples would highlight the effect of the components...

Did the authors carry out TGA-FTIR measurements? Or how do they know that in the first dtep CO and CO2 are forming, especially in nitrogen atmosphere?

A real discussion about the TGA and LOI measurements are also missing.

Regarding the dynamic and sound insulation characterization of tha panel, the rasults can not be understood as they are not compared to any reference. At least the results of a neat epoxy composite should be added. And the 3.5% improvement of the modal damping (line 282) is it a great improvement, or a slight one?

Author Response

The authors express their gratitude to the reviewer for his/her valuable suggestions.

In the revised version of the manuscript, all suggestions have been taken into account. The authors believe that the reviewer comments have identified important areas which needed further improvements and clarifications.

Below, the reviewer will find a point by point description of how each comment has been addressed in the manuscript.

The topic of the paper. the damping, mechanical and flame retardant properties of CFRP is very interesting, however, the manuscript is not suitable for publication in the present form.

Point 1: First of all, the English of the manscript needs to be improved.

Response 1: English has been checked and improved through the manuscript.

Point 2: The introduction is too general, it does not contain any data, and only a short part of it deals with the real content of the ms.

Response 2: Introduction has been implemented referring also to results described in papers on damping assessment of composites, improvement of fire performance and multifunctionality of epoxy-based composites.  Editing and additions are highlighted in yellow.

Point 3:  I suggest adding the informations about the EP curing cycle just after the description of the preparation.

Response 3:

1) Line 175, the text has changed in:  After the sealing, the vacuum bag was transferred into the autoclave for the curing whose pressure and temperature conditions are shown in Figure 2. Figure 3 shows the final manufactured panel.

2) At Line 205 this sentence has been added: The curing cycle which was performed in an oven at atmospheric pressure and was consisting of a 1 h initial step at 125 °C followed by a 3h second step at 180 °C. Vibro-acoustic characterization has been performed on the carbon fiber reinforced composite.

Point 4:  Did the authors detect any sign of filtration of the CNTs during the resin injection process? Is there ny evidence that the CNT content is uniform throughout the composite?

Response 4: in the “Manufacturing of CFR” section line 185 a paragraph has been added: The CFRP panels were prepared with a non-traditional liquid infusion technique. This technique has been already used by Guadagno et al [46] to obtain CFRP panels impregnated using resin with 0.5 wt% CNTs and turned out to be successful to reduce filtration problems and formation of big CNTs agglomerates……..

This paragraph reports  the most important results described in reference 46:  Guadagno, L.; Vietri, U.; Raimondo, M.; Vertuccio, L.; Barra, G.; De Vivo, B.; Lamberti, P.; Spinelli, G.; Tucci, V.; De Nicola, F., et al. Correlation between electrical conductivity and manufacturing processes of nanofilled carbon fiber reinforced composites. Composites Part B: Engineering 2015, 80, 7-14.

Point 5: In the case of the TGA measurements, the method description says that 100C min-1 heating rate was applied, please check it.

Response 5: 100C min-1 has been corrected in 10°C min-1

Point 6:  Please do not use expressions like next picture (e.g. line 204), refere instead to the number of the figure. (see also line 260, table xxx)

Response 6: Line 258 (former line 204) next picture has been modified in Figure 5, line 332 (former line 260) table xxx has been modified in table 3

Point 7: In section 3.1.1 the citation is missing for the mentioned previous work.

Response 7: The missing citation: “Barra, G.; Vertuccio, L.; Vietri, U.; Naddeo, C.; Hadavinia, H.; Guadagno, L. Toughening of epoxy adhesives by combined interaction of carbon nanotubes and silsesquioxanes. Materials 2017, 10” has been added at line 272.

Point 8:  Moreover, the discussion about the double Tg in the case of the GP-CNT sample is not convincing. I suggest the measurement of samples containing only GPOSS, and only CNTs as well, maybe the real explanation will show up.

Response 8: The discussion about the double Tg in case of GP-CNT has been implemented with the support of the literature data.  The curves related to samples containing only GPOSS and only CNT have been added for the discussion

Point 9:  Also in the case of tensile, TGA and LOI measurements, the results of the above mentioned samples would highlight the effect of the components...

Response 9: For Tensile tests and LOI tests the results relative to sample containing only CNT and only GPOSS have been added and discussed. 

Point 10:  Did the authors carry out TGA-FTIR measurements? Or how do they know that in the first dtep CO and CO2 are forming, especially in nitrogen atmosphere?

Response 10: The authors edited the interpretation of the TGA in Nitrogen atmosphere taking account of what it has already reported in literature for epoxy resins.

Point 11:  A real discussion about the TGA and LOI measurements are also missing.

Response 11: TGA in air  and temperature programmed oxidation  (TPO) tests have been added for the neat epoxy sample and for GP-CNT, in order to analyse the combustion products during the thermal scan. Results have been reported and discussed. The discussion about the LOI results has also been implemented.

Point 12: Regarding the dynamic and sound insulation characterization of tha panel, the rasults can not be understood as they are not compared to any reference. At least the results of a neat epoxy composite should be added. And the 3.5% improvement of the modal damping (line 282) is it a great improvement, or a slight one? .

Response 12: The measured damping represents a very huge number for CFRP panel especially if we take into account that the resin does not contain any elastomeric additive that is generally present in most of the standard resin used for these applications. We can imagine as parallelism that nanotubes sample, practically work as a friction damper, while the elastomeric component works as viscous dampers. It has been demonstrated that also a very low percentage of nanotubes can strongly improve the damping factor taking advantage of this effect.

Reviewer 2 Report

This paper deals with the study of CFRP Panel impregnated with an epoxy resin enhanced using a combination of carbon nanotubes and Glycidyl POSS.

The idea is to increase electrical conductivity and flame‐resistance properties.

The paper presents vibro-acoustics tests in order to study the damping and the transmission properties . The experimental results show the efficiency of the proposed formulation.

The experimental process is well described and results are quite new.

In my opinion the paper is acceptable with small improvements, in particular perspectives have to be proposed more clearly (is it possible to imagine modeling ?).

Author Response

Response to Reviewer 2 Comments

The authors express their gratitude to the reviewer for his/her very positive evaluation and his/her valuable suggestions.

Below, the reviewer will find a point by point description of how each comment has been addressed in the manuscript.

This paper deals with the study of CFRP Panel impregnated with an epoxy resin enhanced using a combination of carbon nanotubes and Glycidyl POSS.

The idea is to increase electrical conductivity and flame‐resistance properties.

The paper presents vibro-acoustics tests in order to study the damping and the transmission properties . The experimental results show the efficiency of the proposed formulation.

The experimental process is well described and results are quite new.

Point 1:  In my opinion the paper is acceptable with small improvements, in particular perspectives have to be proposed more clearly (is it possible to imagine modeling ?).

Response 1: The combined effect of nanotubes on stiffness and damping characteristics has being also object of a simulating parallel activity where the specific numerical model for nanotubes integration on resin matrix are under assessment. The random arrangement of nanotubes inside the matrix is a source of uncertainty: for this reason, a stochastic evaluation method seems to be a suitable approach on which the authors are working.

Round 2

Reviewer 1 Report

Thank you for the corrections, in its present form the manuscript is suitable publication.